# Protective factors and sources of support in the workplace as experienced by UK foundation and junior doctors: a qualitative study

Ruth Riley ,[1] Farina Kokab,[2] Marta Buszewicz,[3] Anya Gopfert,[4] Maria Van Hove,[5] Anna K Taylor,[6] Kevin Teoh ,[7] James Martin,[8] Louis Appleby,[9] Carolyn Chew-Graham[10]

► http://dx.doi.org/10.1136/bmjopen-2020-043521

For numbered affiliations see end of article.

**Correspondence to**
Dr Ruth Riley;
r.riley@bham.ac.uk

## ABSTRACT

**Objectives** This paper reports findings identifying foundation and junior doctors' experiences of occupational and psychological protective factors in the workplace and sources of effective support.

**Design** Interpretative, inductive, qualitative study involving in-depth interviews with 21 junior doctor participants. The interviews were audio-recorded, transcribed, anonymised and imported into NVivo V.11 to facilitate data management. Data were analysed using a thematic analysis employing the constant comparative method.

**Setting** National Health Service in the UK.

**Participants** Participants were recruited from junior doctors through social media (eg, the British Medical Association (BMA) junior doctors' Facebook group, Twitter and the mental health research charity websites). A purposive sample of 16 females and 5 males, ethnically diverse, from a range of specialities, across the UK. Junior doctor participants self-identified as having stress, distress, anxiety, depression and suicidal thoughts or having attempted to kill themselves.

**Results** Analysis identified three main themes, with corresponding subthemes relating to protective work factors and facilitators of support: (1) support from work colleagues – help with managing workloads and emotional support; (2) supportive leadership strategies, including feeling valued and accepted, trust and communication, supportive learning environments, challenging stigma and normalising vulnerability; and (3) access to professional support – counselling, cognitive–behavioural therapy and medication through general practitioners, specialist support services for doctors and private therapy.

**Conclusions** Findings show that supportive leadership, effective management practices, peer support and access to appropriate professional support can help mitigate the negative impact of working conditions and cultures experienced by junior doctors. Feeling connected, supported and valued by colleagues and consultants acts as an important buffer against emotional distress despite working under challenging working conditions.

## INTRODUCTION

There are currently 115 376 doctors working in the National Health Service (NHS), almost

## Strengths and limitations of this study

► Few qualitative studies have explored work-related psychological distress in junior doctors; we therefore employed in-depth interviews with 21 junior doctors from across the UK to explore this distress.

► The sample included five male participants compared with 16 female participants, which may be a limitation in this study; however, the purposive sampling of participants ensured the sample was varied in terms of ethnicity, number of years in training, specialty and geographical location.

► Participants self-identified as having chronic stress and/or mental health problems, including anxiety and depression; therefore, the findings may not be generalisable to the general population of junior doctors.

► Semistructured interviews generated rich qualitative data and an iterative process of concurrent data collection and constant comparative analysis facilitated the simultaneous exploration, refinement and enrichment of key themes.

half of whom (56 404) are termed 'junior doctors', which include doctors in specialist training or at preconsultant grade, and foundation year doctors.[1] Previous research indicates that doctors are vulnerable to chronic stress, anxiety, depression and burnout.[2][3] Severe depression has also been identified as a risk factor for suicide.[4][5] High rates of suicide have frequently been reported among doctors but with considerable heterogeneity over time and between countries.[6] Despite mixed findings about suicide rates internationally, there are concerns over suicidal ideation and behaviour in doctors, particularly women.[7][8] International research indicates that psychological distress (such as depression or anxiety) is higher in female doctors compared with male doctors.[9]

The Practitioner Health Programme (PHP) in London, an England-wide specialist mental health, drug and alcohol service for doctors and dentists, reports that two-thirds of those presenting are women, with higher attendance rates among young female doctors.[10] It is uncertain whether this is representative of the general population of doctors or reflects gender differences in help-seeking behaviour.

Most previous research into distress and suicide in junior doctors reports quantitative, retrospective studies, with the focus on the individual risk factors (particularly emphasising the need for 'resilience'),[11 12] and overlooking the wider systemic, organisational and cultural sources of distress among this population of health professionals. Focusing on individual risk factors and interventions are likely to detract from considering solutions or interventions targeting the learning environment, organisational culture and systemic factors.[13 14] To our knowledge, no previous study has qualitatively examined how junior doctors view their working conditions (eg, rotas, spaces to rest, and eat and sleep when on-call) and work cultures (relationships, support and job control) and the factors (eg, debriefs following critical incidents, policies, available support and management practices) that may protect them from psychological distress or offer them support.

Junior doctors face specific pressures related to their professional stage and development. In a nationwide survey of junior doctors in Ireland, stress, anxiety and depression were highest among junior doctors as compared with consultants.[15] A study exploring junior doctors' reasons for leaving medicine in the UK cited lack of support, mentorship or formal training, loneliness and bullying.[16] A qualitative study undertaken in Australia found that lack of support from colleagues, working beyond one's perceived abilities and neglecting self-care were key contributors to burnout and reduced well-being among junior doctors.[17] Doctors under investigation are also at higher risk of suicide and psychological distress[18 19]; those in receipt or facing threats of complaints experience elevated anxiety, depression and suicidal thoughts compared with colleagues who face no complaints.[20 21]

A lack of support, mental health stigma and toxic work cultures characterised by bullying and a blame and shame culture were identified as a key source of distress among junior doctors, in a separate paper linked to this study.[22] Similarly, feeling undervalued, unsupported or having reduced autonomy were key factors affecting low morale among junior doctors[23]; this highlights the impact of working conditions and work culture on junior doctors and the potential value of providing effective support and supervision.

There is substantial evidence linking longer working hours with mental ill health and reduced patient safety; junior doctors who worked over 55 hours a week were more than twice as likely to report a common mental health problem[24] and increased likelihood of making a medical error.[25] This highlights the importance of providing suitable facilities for sleep, rest and social spaces and to facilitate social interaction and access to collegial support.[26]

Evidence has found that junior doctors valued the emotional support derived from attending debriefing sessions following adverse events, although its effectiveness in reducing burnout was equivocal.[27] Workplace interventions such as Schwartz rounds, which provide permission and a safe space for healthcare staff to talk about the range of feelings and challenges that arise during their work while also fostering connectedness to others, were valued by staff.[28] Although numerous workplace health theories[29 30] identify effective support as a core antecedent to positive mental health and as a potential buffer against challenging work environments, how this might manifest among junior doctors is unclear. Identifying such patterns or contextual factors is crucial in developing targeted improvements or interventions that can reduce the risk of mental ill health and psychological distress from the outset.

Occupational or work-related stress are also compounded by barriers to help-seeking that include difficulties accessing healthcare due to working hours, perceived stigma, concerns about loss of confidentiality, therapeutic nihilism (a cynicism about the effectiveness and availability of existing interventions for mental ill health) and pressure to continue working while unwell due to staff shortages and concerns about letting colleagues down.[2 9 31 32]

Facilitators included the availability of confidential, specialised services for doctors, and emotional and practical support such as finding cover for shifts.[31] Shanafelt et al[33] recommend various interventions (eg, positive role models, training of group and teams and interprofessional support groups) to facilitate culture change. Promoting well-being by providing psychological safety and positive learning environments may reduce learning anxiety among junior doctors.[34]

This qualitative study is part of a wider mixed-methods study examining the psychological, cultural and occupational contexts associated with reduced psychological well-being in junior doctors and protective factors that may mitigate against this. This paper reports the qualitative findings focused on the protective factors and sources of effective support in the workplace as experienced by junior doctors.

## METHODS

### Study design and setting

The study methodology was underpinned by a constructivist grounded theory approach,[35] using qualitative methods, with semistructured interviews to explore junior doctors' perspectives and experiences of stress and distress. The study setting was the NHS in the UK.

### Sampling and recruitment

We used a range of recruitment methods including advertisements on social media (eg, the junior doctors'

## Box 1    Inclusion and exclusion criteria

**Inclusion criteria**
► Currently, a foundation or junior (preconsultant) doctor working in a hospital in the National Health Service (NHS) in England/Wales within the last 2 years – this covers any preconsultant doctor working in the NHS. They do not necessarily have to be on a training contract.
► Experience (within the last 4 years) experience of stress, distress, mental illness, self-harm (eg, cutting, overdoses), suicidal thoughts, feelings and intent.
► Has capacity to provide informed consent (this will be assumed given they are healthcare staff).

**Exclusion criteria**
► Currently experiencing acute severe mental illness such as psychosis.
► Currently in receipt of drug and alcohol services (this does not include doctors who currently use alcohol/drugs to self-medicate such as doctors who may be using alcohol/drugs on a regular basis to help them relax but would not be classified as having an addiction problem).
► Actively suicidal or anyone who has made a suicide attempt within the last 6 weeks – to avoid causing any additional psychological harm during a period of acute distress/vulnerability.

## Box 2    Interview topic guide

**Introduction and background**
► Describe general less stressful/more stressful jobs and the difference between these.

**Work environment**
► Describe main sources of stress in day-to-day working life.
► Explore wider sources or stress not already mentioned.
► Explore which jobs are more stressful.

**Impact of work on mental health and well-being**
► Past, present and future outlook as to how work impacted/may impact mental health and well-being?

**Preventing/seeking/managing help**
► Explore management of workload/stress in day-to-day work life.
► Discuss help-seeking for distressing events.
► Explore relationships with colleagues and how concerns are responded to if raised.

**Experience of help-seeking**
► Explore thoughts/feelings of seeking help.
► Explore knowledge of available support and protective factors.

**What could make things better?**
► Explore any realistic changes at individual and organisational level and upstream changes.

Facebook group, Twitter and the mental health research charity websites). In addition, information about the study was circulated through the PHP, a specialised mental health and drug and alcohol service dedicated to doctors and dentists.[36]

Potential participants were asked to express their interest by contacting members of the study team who then provided further information about the study.

The research team employed purposive sampling from the responding junior doctors to ensure maximum variation taking account of the following characteristics: gender, age, ethnicity, sexuality, geographical location, different grades/duration of General Medical Council registration, medical specialty, disclosure of a mental health diagnosis, individuals who reported self-harm behaviour, had thoughts of suicide or had attempted to kill themselves.

The eligibility criteria used are listed in box 1.

Junior doctors interested in taking part were sent a reply slip and consent form electronically and asked to sign the consent form prior to the interview, before returning it to the study researcher. Face-to-face, telephone or 'virtual platform' interviews were arranged at a time convenient to each participant. They were given the opportunity to raise questions prior to being interviewed and during the interview.

At the end of each interview, the researcher informed the participant about the next steps, how their data would be used and checked on their well-being prior to leaving the interview setting. A risk protocol was used to ensure appropriate support was provided to participants (and the researcher) in the event of the disclosure of significant distress or suicidal ideation.

### Data collection
A topic guide (see box 2) was developed by the research team to generate discussion in the semistructured interviews. The topic guide was informed by the existing literature, input from junior doctors on the study team and patient and public involvement input (see further) and modified iteratively as data collection and analysis progressed. The topic guide aimed to capture participants' views, experiences, feelings and beliefs about working conditions and cultures that were perceived to be stressful or distressing and factors that were perceived/felt to be protective.

One-to-one interviews were conducted face to face (n=7), by telephone (n=13) or using a digital virtual platform (n=1) and digitally recorded with consent. The in-depth interviews were conducted by two researchers (FK and RR), both female social and behavioural scientists with qualitative methods expertise. The recorded interviews were transcribed verbatim and checked for accuracy of transcription by the study researcher before analysis. All transcripts were anonymised before discussion within the wider research team. Reflexive notes were recorded throughout the process. Recruitment and data collection were continued until data saturation was judged to have been achieved.[37]

### Data analysis
Analysis began with data collection and was iterative and inductive, employing the constant comparative method[38 39] until theoretical data saturation was achieved, such that no new analytic categories emerged.[37]

The research team included three junior doctors (AG, AKT and MVH), two academic general practitioners (CC-G and MB), one occupational health psychologist (KT) and two social scientists (RR and FK). FK coded the data set; RR coded a subsample and contributed to the organisation of themes. The multidisciplinary team provided commentary on transcripts to generate and refine codes and thematic categories and provide researcher triangulation, thereby increasing the credibility of the research findings.[40] Data were managed using NVivo V.11. The study is reported in line with the Consolidated criteria for Reporting Qualitative research.[41]

### Patient and public involvement

There were three junior doctors on the research team, all of whom consulted with colleagues about the initial research idea. It was felt important that the research focus needed to explore working conditions and cultures, rather than focusing the gaze inward or on individuals. Five junior doctors provided feedback on the initial funding application, and four of these gave feedback on the protocol, topic guide and participant facing documents. Junior doctors have been actively involved in publicising the study to junior doctors, and we are working on a dissemination strategy with junior doctors outside the research team. Due to the time constraints of junior doctors, Patient and Participant Involvement and Engagement (PPIE) members were consulted via email and telephone.

### FINDINGS

Twenty-one interviews were conducted, lasting between 43 and 103 min (mean=65 min), between November 2019 and May 2020. The demographic and professional characteristics of participants are included in table 1. Analysis of the interview transcripts and field notes identified three main themes, with corresponding subthemes relating to protective work factors and facilitators of support: (1) support from work colleagues, including help with managing workloads and emotional support; (2) supportive leadership strategies, including feeling valued and accepted, trust and communication, supportive learning environments and challenging stigma and normalising vulnerability; (3) access to professional support, including cognitive–behavioural therapy, counselling and medication through general practitioner and specialist support services for doctors and accessing private counselling/therapy.

### SUPPORT FROM WORK COLLEAGUES

Many participants emphasised the importance of working in protective and supportive work cultures, characterised by good interpersonal relationships, a strong team morale where there was a shared responsibility for workloads and where staff felt supported with the clinical components of their work and the emotional impact of their job.

**Table 1** Participant characteristics

| Participant characteristics | n=21 | % |
|---|---|---|
| Gender (female) | 16 | 76 |
| Age (years) | | |
| 20–29 | 10 | 48 |
| 30–39 | 11 | 52 |
| Ethnicity | | |
| Asian (other) | 2 | 10 |
| Bangladeshi | 1 | 5 |
| Chinese | 1 | 5 |
| Indian | 3 | 14 |
| White | 13 | 62 |
| White (other) | 1 | 5 |
| Sexual orientation (heterosexual) | 15 | |
| Years since qualification | | |
| 0–5 | 10 | 48 |
| 6–10 | 9 | 42 |
| 11–15 | 2 | 10 |
| Specialty | | |
| Emergency medicine | 2 | 10 |
| Medicine (including acute, diabetes/endocrinology, geriatrics) | 9 | 42 |
| Obstetrics and gynaecology | 6 | 28 |
| Paediatrics | 2 | 10 |
| Psychiatry | 1 | 5 |
| Public health | 1 | 5 |

Crucially, participants highlighted the interdependence between collegial support, supportive work cultures and their emotional well-being and ability to cope with the demands of their job.

(a) Help with managing workloads

Participants identified the benefits of having mutually supportive interpersonal connections at work where there was a willingness to share responsibilities, offer practical support and helping with workloads or specific tasks:

> I've made good relationships… [name of speciality] was actually having a very, very nice group of colleagues on the whole and supporting each other. Like offering help if we're struggling, if someone's struggling with a task then I would offer help and vice versa. (JD12, male)

The following participant highlighted that nurses offered support by undertaking tasks that free up time for junior doctors to focus on clinical work: spending time with supportive staff could alleviate some of the burden and stress that junior doctors may be experiencing, especially in new and unfamiliar roles.

> My relationships with my colleagues, both nurses and doctors, are incredible. I couldn't say a bad word

about any of them. They have been so wonderfully supportive. The nursing staff have carried me in every job I've been on. They've carried me because they're the constant and they're there all the time. (JD17, female)

(b) Emotional support

Participants iterated the importance of peer support and feeling emotionally supported within the team, for example, in response to critical incidents, or being bullied by a colleague:

A couple of weeks ago my friend was having a terrible time with a registrar who was being hideous to her and so, you know we kind of helped support her through that and empathise with her and you know I think that's really helpful. (JD20, female)

The following participant referred to the '*in it together*' team spirit and camaraderie that acted as an emotional buffer when working under challenging working conditions:

We were all in it together and actually, the working culture is really, really good … even in the rubbish conditions that we're working in, it's the people that have made it worthwhile and they've helped me cope with things, so if I'm having a bad day or I'm a bit down, actually it's the camaraderie of working with midwives, other doctors, nurses, pharmacists, porters that have made it quite a happy thing to do really. (JD09, female)

Many participants had contrasting experiences on different rotations; the following participant described her experience of being in a supportive work culture in the context of a critical incident, which she compared with the 'blame culture' experienced on a previous rotation:

In the unit I'm in at the moment, it's a lot more about learning rather than blaming. I think you feel safer then to express opinions or to discuss cases. You want to find out who was involved in order to be able to support them and that's the feeling that you get from the senior midwives and the consultants you work with… The unit that I was in before was really challenging in terms of feeling like you wouldn't be supported, even if you'd done nothing wrong. (JD13, female)

Another participant reported feeling '*emotionally healthy*', which she attributed to the supportive work culture. She contrasted this with a previous rotation where the cultural norm was to disengage with challenging emotions, leaving staff with unprocessed trauma:

I feel really emotionally healthy now. Cause I'm in such a supportive culture. But I look now particularly at [name of specialty] culture as a lot of people who, with a lot of unprocessed trauma. Huge amounts of unprocessed trauma. Amazing really that whole profession can be so in denial of the emotional impact of their job. (JD23, female)

The value of working in a supportive work culture was further highlighted by the following participant who recollected that having greater support from colleagues in her previous post would have reduced her sense of anxiety:

I think if I'd have had that [informal support from colleagues] support in first year of someone who could listen, maybe I wouldn't have been so anxious about everything. (JD09, female)

The following participant suggested that, despite the potential stigma associated with mental illness, peer support was often the first line of support for colleagues:

I do think peer support is very important and I think that's probably the first line for a lot of people, like even with some stigma and embarrassment about mental health, I think most people would usually turn to a colleague first. (JD16, female)

## SUPPORTIVE LEADERSHIP STRATEGIES
### (a) Feeling valued and accepted

This reoccurring theme refers to supportive and proactive leadership and management approaches to providing support to the team and promoting positive mental health. Participants relayed the importance of feeling valued, included and accepted as part of the team by the consultants. This included receiving recognition and praise for their efforts and contributions, particularly when working in challenging, highly pressurised environments and working long shifts:

… it [first job] was over 12 hours every single day, you know I think I started to cry because just at least someone was acknowledging the fact that we were doing 12, 13 hour days every day … it was extremely hard but…there was a very good group of consultants who just acknowledged that it was hard, there was not much they could do about it, they had to work hard as well but they were very, very, very supportive of that… they [consultants] just knew what your name was, they knew your role, you felt valued there. (JD01, male)

… it makes such a difference and then also like being empowered to be of good patient care, so you know not being shouted at if you've taken half an hour to see a patient and actually you know being recognised, 'Oh you did a really thorough job actually, well done, good for you' and kind of you know just actually some recognition of what you're doing. Yeah, that's what is good, I guess. (JD20, female)

This participant emphasised the importance of feeling valued by consultants or patients and its relationship to job satisfaction and confidence:

… when you receive some good feedback, you can really get a bit of a buzz and a bit of a high from that,

either from a patient or from a colleague when they said, you know, 'you've really made a difference today' or 'you've really helped me today' or 'you've made me feel better, thank you'. Whenever you get something like that, which doesn't happen all the time by any means, but when you get a bit of feedback, it makes you realise why you're doing it and I guess all those feelings of anxiety and the feeling of not being good enough, which is a big thing for me. So, if somebody's appreciative, I think that can brighten your day really and make you feel like you're worth something. (JD09, female)

(b) Trust and communication

A few participants recalled the value of consultants who were proactive in their support of staff, openly inviting them to discuss concerns, ask questions or to enquire about their work and well-being. Establishing these open channels of communication and being approachable made it easier for participants to seek support or advice when needed:

Whereas, the two consultants that I go to for that sort of support always made it clear that they were open to talking about anything that was bothering me. They continued to keep that relationship open, even when I wasn't working with them anymore. It was just the odd touch base email like, 'How are things going? What are you up to? Let's have a coffee one day'. It was that kind of thing. You're made to feel a bit less like you're onerous or that you're bothering them when you ask them about something. (JD13, female)

Such was the importance of supportive relationship with consultants, a few participants had maintained those connections after moving to a new post:

I've got a couple of people that I go back to when I need support ….and I know will speak to them both but I will always go back to those few people that I trust and I know. (JD19, female)

(c) Supportive learning environments

Participants reflected on work cultures where there was a shared commitment to team learning. Such an ethos enabled people to practice without fear of reprisal, to reflect on their work and to view errors as opportunities for learning and improvement:

My ST1 [Specialist Training year one] year was extremely enjoyable, because again it was a very supportive place to work and they encouraged you to ask questions and didn't criticise and there was a real sort of no blame culture. (JD01, male)

Some participants suggested that cohesiveness within teams could be enabled through multidisciplinary training opportunities. These events facilitated a shared understanding of different roles within the team and awareness of the challenges and concerns faced by different professional groups:

Where I am at the moment is really good. The current unit I'm in has a lot of multi-professional training that isn't necessarily clinical. We have multi-professional leadership days where the senior junior doctors at ST5 level plus and the senior midwives all get together to do the training together. I think that really helps because we understood their role and their concerns when they're coordinating a unit and they understand that we can't really do everything. (JD13, female)

(d) Challenging stigma and normalising vulnerability

A number of participants emphasised the role of consultants and supervisors in destigmatising mental ill health by sharing or disclosing their vulnerability or experiences of mental ill health to their staff; participants report that this created a more open and authentic work culture where staff also felt safe to disclose their vulnerability:

We were working a Sunday together and I said, 'I haven't seen you in a while.' She said, 'No, I have been off for about four months.' I said, 'Are you just coming back?' She said, 'Yeah, I was off because I had a mental breakdown. One day, I just realised that I was really, really unwell and I just had to go home and I didn't return for a number of months until I got myself sorted out.' My point was just her talking about that, I think, does wonders because she's a woman who has been there forever. She's a senior consultant. Just telling other people, 'Yeah, I've gone through the struggles. I need medication for this, that and the other. It's okay. I'm still here functioning and it's fine'. I think people need to be more open about that. (JD03, male)

I think what really needs to change is that I think that more senior doctors need to be open about their struggles and their issues in training because I think there's a perception that you need to be invulnerable among more junior trainees in particular, which I think as people gradually have breakdowns, they're robbed of that illusion. Yes, yes, I think that kind of cultural change needs to kind of happen. (JD07, female)

## ACCESS TO PROFESSIONAL SUPPORT

Few participants reported seeking formalised help from within the workplace (eg, occupational health) but some study participants had sought professional help from counsellors, therapists, general practitioners (GPs) and specialist services for doctors, outside the workplace. The following participant was prescribed antidepressants by her GP to manage her distress:

… I was in tears and nervous … I kept on saying, 'I don't feel like this is going to ever change. I don't feel confident,' and then I started antidepressants and incredibly, within about two weeks, I was sleeping. I felt

better and I felt calm. I realised just how sick I had been. (JD08, female)

Accessing external support was not without barriers. The following participant highlights the significant delay in accessing a counselling referral through their GP:

My GP referred me for local support and it took them a year to contact me… I mean, it's just the NHS, that's how it is… and even then, it would be one o'clock, two o'clock, it would be nigh on impossible to actually access. (JD19, female)

Having access to services tailored to the needs of junior doctors was important for participants:

Because you know that their [specialist service] whole patient group are doctors and they've got such expertise of talking to doctors and professionals with the same issues, there is none of that having to make an excuse for yourself. We're so often held up in society as a really responsible, clever and respected group that you can be vulnerable in the PHP. That was really, really validating. (JD08, female)

The following participant found out about the PHP by chance (a family member found a PHP leaflet in a waiting room); she was concerned about being referred to a psychiatrist in her local trust because she was planning on undertaking a rotation there in the future:

There is the Practitioner Health Programme… I've been seeing a GP, that lives down the road from me, who's absolutely amazing… I feel like that's a really good setup… I didn't feel comfortable being treated in the mental health trust here and also, I want to look at working in that mental health trust. Being able to move my care completely away from that is very important. I didn't know that was there at the beginning. That's when I ran into some of my problems and when I wasn't seeking the help I needed because I didn't know that I could have anonymity. (JD22, female)

Due to concerns about the potential loss of confidentiality, the following participant self-referred to a private clinic to seek the formal support they needed:

I went to a private mental health clinic and I referred myself, I phoned up and (? they) said well the Consultant probably would take a self-referral especially if you're a Doctor, so I wrote my own referral letter and I went privately to the Psychiatrist across the road from the hospital and he got me some treatment which was mainly in the form of counselling … And that changed everything, you know that turned everything around. (JD15, female)

A few participants felt it would be beneficial if junior doctors had access to monthly therapy or supervision sessions, either group work or one to one, a safe space to share and process the emotional impact of the job and seek support:

If, as junior doctors, you saw a therapist once a month… if we made it universal (which is very easy to say) that for one hour a month, everyone had to see someone to talk to, I think that would be a good way of moving forward. If there was something that was really serious, it's a safe space and we can talk about it. If it needs to be escalated or if any sort of treatment needs to be started in whichever way that might be, go for it. (JD03, male)

… sometimes lots of stuff coming up and yet there's absolutely nothing. For me I feel like, every doctor should have some form of emotional supervision. (JD23, female)

## DISCUSSION

The findings highlight the value of support from work colleagues and work cultures engendered by camaraderie and connectedness with colleagues. The emotional support offered by colleagues often enabled junior doctors to manage the challenges of the job and was considered to be potentially protective against mental ill health. Participants felt more able to cope with the challenges of their work when their managers and leaders valued staff, offering clinical and emotional support and creating positive learning environments that reduce learning anxiety by providing psychological safety.[33][34] Positive learning cultures, characterised by social support, validation of success and positive affirmation have been found to be protective against burnout.[42] These findings also support the job demand–resources theory[29]; namely, that key resources such as support from colleagues, quality of the relationships with supervisors/consultants and feeling valued can buffer the impact of those demands related to their work (long hours, high workload) and consequently protect against burnout.

Our findings also highlight the role of consultants and other senior staff in destigmatising mental ill health by acknowledging and sharing their own vulnerability and creating a safe and authentic work culture that enabled staff to seek support. Promoting emotionally open cultures that facilitate disclosure by destigmatising mental ill health and allowing for vulnerability empowers staff to seek support and could reduce the sense of isolation and shame often experienced by junior doctors.[31] Challenging and addressing stigma in medical school and beyond is a crucial stepping stone in enabling doctors to seek help without fear of reprisal or shame.[9][43][44]

These findings highlight the positive impact of harnessing existing workforce capital (ie, the skills and training of staff) to sustain cultures, practices and leadership styles that promote and engender supportive working environments and relationships. Participants in this study valued the support provided by work colleagues; this is underpinned by evidence attesting the value and

importance of social relationships at work in buffering the demands of the job.[45] Belonging, feeling valued and connectedness are protective factors and can promote positive mental health and a positive professional identity; for instance, peer/colleague support in the workplace has been found to mitigate the negative psychological consequences of adverse events.[45 46] Authors recommend checking-in with colleagues, listening, reflecting, reframing, sense-making, coping, closing and resources/referrals that can reduce the isolation and sense of shame experienced by colleagues.[47]

Our findings highlight the value of working in supportive, cohesive and connected teams and are supported by previous research that found that staff who work in cohesive teams have better mental health, while lack of leadership was associated with poorer mental health.[48 49] Having continuity in teams, including a named consultant fosters a sense of belonging and produces better patient outcomes.[49 50]

Participants in this study highlighted that despite challenging working conditions—long hours, high workloads—they felt better to be able to cope with these demands and less isolated when they operated within a supportive work culture. Such cultures were often enabled and enacted by good leadership; consultants with proactive leadership and management styles that allow staff to feel supported by being valued, involved, heard and respected. Consultants who recognise and value the skills, knowledge and experience of team members, invite staff to ask questions and advertise their availability can facilitate learning and empower staff to ask questions or for help, if needed. Such examples of effective management and leadership are supported by evidence that has shown that effective leadership can promote mental health and well-being, prevent stress and enable staff to perform at their optimum.[51]

Training for leadership styles and strategies that promote team cohesion or which, conversely, reduce the antecedents of toxic work cultures, including bullying, role conflict, role ambiguity, and chaotic and unpredictable work environments,[52] is crucial. Employers also need to place greater value on meaningful, inclusive and impactful leadership and management styles that support staff.[53] However, NHS policy makers must also ensure that efforts are made to deliver sustainable workforce cultures that avoid an overdependency on individual leaders. A whole systems approach to prevention and provision will need to be adopted to promote healthy workplaces for all NHS staff. This includes addressing systemic causes of stress such as workforce shortages, rota gaps and improving physical working conditions while also promoting work cultures that enable help-seeking and foster belonging and connectedness.[13]

Working in effective and supportive teams requires staff to have shared objectives, values and continuity, necessitating planning and changes to rotas that can help to ensure junior doctors feel part of a team.[54] There is substantial evidence that working in teams that lack continuity in staff or shared objectives is detrimental to the mental health and well-being of staff and impacts negatively on patient care and safety.[49 50] Junior doctors need to feel valued, capitalising on existing support infrastructure and assets (ie, teams, supervisors and colleagues) to provide clinical and psychological support (eg, interprofessional debriefs for processing and validating feelings). Organisations will need to provide the training and coaching to develop effective and cohesive teams, while also meeting recommendations set out in existing guidance to ensure the clinical, learning, training and emotional needs of doctors are met.[55]

Findings from this study, reported in a sister paper,[22] revealed the potential for emotional distress when junior doctors work in contrasting cultures where there are no safe spaces to discuss difficult or traumatic work experiences, such as the sudden or violent death of a patient. Feeling supported, permitting time out or time off work when needed and minimising any potential for isolation and distress in such circumstances can reduce the likelihood of secondary/vicarious trauma (a process of change resulting from the effects of prolonged exposure to human suffering and trauma).[56 57] Feeling isolated when distressed is also a risk factor for suicide[58]; providing effective support needs to be an imperative in such contexts.

The provision and access to confidential support was important to participants and evidenced by previous research.[2 31] Being able to find a safe, secure, anonymous and easy-to-access source of support outside of their work environment was greatly valued. To ensure doctors are aware of the available support, signposting information about specialist services such as the PHP could be included in every induction session across the country and be reinforced by specialist training schools through Health Education England. Some participants however preferred accessing a trusted GP. Managers and leaders therefore need to facilitate and sanction appropriate support for staff when needed.

Participants' experiences in this study are supported by the findings of a realist review of interventions and strategies to reduce mental ill health and distress in doctors. This found that leadership and supervision level interventions, supported by the organisation, focusing on relationships, emotional support and belonging were more likely to promote well-being and that fair and equitable learning environments that balanced positive/negative feedback enabled doctors to thrive.[59] This is underpinned by evidence that found that social connectedness contributes to the reduction of stress levels while social identification (belonging) is beneficial to the psychological health of employees and reduces burnout.[45 54 59]

Leadership and management can play a key role in shaping group identities and belonging within a team.[60] Trust, defined as 'the extent to which one is willing to ascribe good intentions to, and have confidence in, the words and actions of other people'[61] has been shown to be positively associated with performance, civility and psychological safety.[62] Workforce cultures and environments play

an integral role in promoting the mental health of junior doctors through support and creating safe and authentic environments where doctors can excel in their training, and overcome challenges linked to the demands of their job, enabling them to stay well.

This study has some limitations. There is a notable disparity in gender, with a higher proportion of female doctors participating. We received higher expressions of interest from female (n=37) participants compared with male (n=22) participants, with a higher proportion of female participants then agreeing to participate in an interview. The interest in this study and increased willingness among female participants to come forward and talk about their experiences may reflect evidence indicating that female doctors are more likely to experience distress, with increased rates of suicide evidenced in young female doctors. The higher proportion of female participants may also reflect gendered help-seeking behaviour for mental ill health, evidenced also in the wider population.[63] Another potential limitation relates to the lower number of participants who reported having had suicidal thoughts (n=5) and participants who disclosed that they had made self-harm attempts (n=2).

## CONCLUSION

Our findings show that support from colleagues, effective and supportive leadership and management practices and access to appropriate professional care can help mitigate the negative impact of working conditions and cultures experienced by many junior doctors. Feeling connected, supported and valued by colleagues and consultants acts as an important buffer against emotional distress despite working under challenging working conditions and reduces any potential for isolation.

Authenticity of interactions between senior and junior colleagues was seen as an important aspect of how mental health and well-being are understood and negotiated in the work environment. Normalising vulnerability through disclosure and creating emotionally open cultures where vulnerability is accepted and understood allows junior staff greater confidence to be open about factors affecting their own well-being and to seek and receive support when needed. Supporting doctors who request time out or time off and facilitating access to support could reduce the potential for isolation in the workplace and reduce stigma-related barriers to help-seeking. Examples of effective interventions and solutions to minimise distress and support staff are evidenced in existing leadership and collegial support but need to be more consistently practised across the NHS.

**Author affiliations**
[1]College of Medical and Dental Sciences, University of Birmingham, Birmingham, UK
[2]School of Social Policy, University of Birmingham, Birmingham, UK
[3]Research Department of Primary Care and Population Health, University College London, London, UK
[4]Oxford University Hospitals NHS Trust, Oxford, UK
[5]London School of Hygiene & Tropical Medicine, London, UK
[6]Faculty of Medicine and Health, Leeds Institute of Health Sciences, Leeds, UK
[7]Department of Organizational Psychology, Birkbeck University of London, London, UK
[8]Institute of Applied Health Research, University of Birmingham, Birmingham, UK
[9]Department of Psychiatry & Behavioral Sciences, University of Manchester, Manchester, UK
[10]School of Medicine, Keele University, Keele, UK

**Acknowledgements** The authors would like to thank all volunteer junior doctor participants who kindly gave their time to share their experiences of working in the National Health Service. We also wish to thank members of the PPIE group who provided valuable input throughout the study.

**Contributors** RR, CC-G, MB, KT, AG, AKT, MVH, LA and JM: substantial contributions to conception and design. RR, FK, CC-G, MB, KT, AG, AKT, LA and MVH: acquisition of data or analysis and interpretation of data; drafting the article or revising it critically for important intellectual content; and final approval of the version to be published.

**Funding** The study was funded by NIHR Research for Patient Benefit (PB-PG-0418-20023).

**Competing interests** None declared.

**Patient consent for publication** Not required.

**Ethics approval** Ethical approval was granted by the University of Birmingham and Health Research Authority (reference number: 19/HRA/6579).

**Provenance and peer review** Not commissioned; externally peer reviewed.

**Data availability statement** All data relevant to the study are included in the article. This study has not received ethical approval to share confidential data with any third party other than the study research team.

**ORCID iDs**
Ruth Riley http://orcid.org/0000-0001-8774-5344
Kevin Teoh http://orcid.org/0000-0002-6490-8208

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
