## [Reviewer comments · BMJ Open]

ARTICLE DETAILS

TITLE (PROVISIONAL)	Protective factors and sources of support in the workplace as experienced by UK foundation and junior doctors: a qualitative study
AUTHORS	Riley, Ruth; Kokab, Farina; Buszewicz, Marta; Gopfert, Anya; Van Hove, Maria; Taylor, Anna; Teoh, Kevin; Martin, James; Appleby, Louis; Chew-Graham, Carolyn

VERSION 1 – REVIEW

REVIEWER	Miharu Nakanishi Tokyo Metropolitan Institute of Medical Science, Japan
REVIEW RETURNED	10-Nov-2020

GENERAL COMMENTS	The present study examined protective factors and sources of support in workplace among foundation and junior doctors who experienced stress, distress, anxiety, depression and suicidal thoughts. The authors clearly summarised the core components from narratives. It would be recommended to discuss possible facilitators and barriers in NHS - as this study emphasised the scarce evidence in organisational culture and systemic factors - to establish peer support and supportive leadership strategies in the workplace. As participants mentioned in the narratives, these factors may have varied across different workplaces. It thus seems to depend on some personal attributes of leaders whether the workplace can establish supportive culture and strategies. Does NHS have had specific strategy to reduce such dependency on individual leader's skills? Further discussion would be helpful for international readers to provide implications for healthy workplace for young doctors. Discussion on access to professional support, with citation of PHP, would also be beneficial. Participants stated that they preferred such services rather than mental health services at local trust, because of accessibility or their concerns about being acknowledged by possible colleagues in the future. The latter would be worthy o
---

REVIEWER	Andrea Cioffi Sapienza University of Rome, Italy
REVIEW RETURNED	04-Dec-2020

GENERAL COMMENTS	Interesting article. The topic chosen is extremely topical and the typical problems of junior doctors are particularly important today because they influence the performance of the National Health
--

	Systems. The Manuscript is well structured and methodologically correct. It is advisable to report the first part of the findings section ("Twenty-one interviews were conducted") in the methods section. This because the research methods are described: number of interviews, interview time, questionnaire structure (3 themes).
--	--

REVIEWER	Margaret Kay University of Queensland, Australia
REVIEW RETURNED	13-Dec-2020

GENERAL COMMENTS	Thank you for the opportunity to review this paper "Protective factors and sources of support in the workplace as experienced by UK foundation and junior doctors: a qualitative study." The paper is well written and easy to read providing important practical understandings regarding the protective factors and sources of support for junior doctors. The Abstract seems adequate as a summary of the article, though it would be helpful to have a little more information about the participants/recruitment here if possible. In the Strengths and Limitations section, the authors could include the fact that the junior doctors who were recruited were doctors who had experienced mental health issues. This is significant because these doctors will visualise the issues through that lens and this is likely to impact their reporting of the issues being researched. It potentially limits the transferability of the findings across the junior doctor population. It can also be viewed as a strength. Either way it needs to be identified as something that could impact the findings. The authors mention that they included many more women than men. While there were more women compared to men who were potential participants (37:22), the ratio of women to men in the actual participants after purposive sampling included a higher proportion of women (16:5). Given that gender was one of the issues considered in the purposive sampling, it is not really clear why there needed to be such significant weight given to including women. The justification in the Strengths and Limitations section is not adequate and this should be addressed clearly in the paper – perhaps in the methods section. The introduction is a succinct description of the current literature that has a focus on the UK/Irish literature and a broader review of the literature is relevant here. The study is internationally relevant. The work of Shanafelt et al at Mayo Clinic on workplace culture and organisational aspects of doctor wellness could be included. While the authors are correct that there is a paucity of literature on junior doctors, there are studies, including Hoffman and Bonney (2018) "Junior doctors, burnout and wellbeing et al." that reports findings that are relevant to this current study. While studies may focus on barriers, they may still report on findings about supportive work practices and engaging with the broader literature would strengthen this paper. Searching the literature again and including such papers would strengthen the introduction significantly. On page 7 – lines 53-56 – the sentence, "Identifying such patterns or contextual factors is crucial in developing targeted improvements or interventions that can reduce the risk of mental ill health and psychological distress from the outset." Is unclear in its meaning. Perhaps the authors could provide develop this statement to provide more detail for the reader who may not be as familiar with
--

the workplace theories. It would be helpful to then refer to the more relevant theory/theories in their discussion to round out this concept. Currently the discussion seems to refer to this theory in the phrasing of the concepts but this is not overtly stated. The references in this introduction seem to be to the job-demand-resources theory its variations. Is this a key theory that the authors wish to highlight? Do they feel that their data confirm this as a useful theoretical approach to understanding their findings, or do they have mixed feelings about how their data fit this theory? Currently the introduction of the concept of workplace theories, leaves the reader hanging to hear the authors' conclusion.

The Method section – see comments above about the gender ratio. The definition of PPIE members was not included in the methods and it would be worth clarifying for the reader how the PPIE members could and did inform this study about junior doctors in more detail. It is possible that this contribution was significant and it is of interest to the reader. If it was not a significant contribution, then understanding how to enhance that contribution into the future for other researchers is equally important. Doctors' health and wellbeing is an important issue for our community as a whole given the association between maintaining the wellbeing of doctors and the delivery of quality care.

In the Method section, it seems to be implied that the analysis was inductive, but this needs to be clarified by the authors. If this was not the case, more detail would be helpful. It would also be useful to know the number of face-to-face and the number of the other type of interviews that were undertaken.

In the results section, the definition of peer support seems quite broad. It seems that the quotes include the support from nurses and other team members, not just peers. The authors may wish to reconsider the title of this key theme that better encapsulates this breadth of support received by the junior doctors e.g. such as support by work colleagues.

The Discussion is well presented. (see notes about the Introduction above regarding the introduction of the workplace theory). It would benefit from a deeper engagement with the international literature on this topic, especially the other qualitative studies. There are also advantages to situating the qualitative data within the known quantitative data about junior doctors. Currently the discussion seems to focus on NHS and healthcare management references rather than references to the doctors' health literature. The findings are reinforced as strong statements but should be strengthened with reference to the existing knowledge about doctors' health.

On page 15, the abbreviation of PHP is used in line 21 before it is explained in line 23. The PHP is mentioned in the discussion. It is not clear if the authors think that this is the best option, rather than a closer GP? Accessing health care with a trusted GP is likely to be a reasonable option and perhaps this should be highlighted in the discussion as well?

The discussion includes some recommendations to address the concerns raised, but they seem to be quite vague concepts. Is it possible for the authors to give more specific recommendations?

The references seem appropriate, but predominantly have a narrow focus on UK literature. A number of references, including 7, 10, 11, 12, 13, 19, 23, 26, 28, 35, 37, seems incomplete.

Overall, I think this is an excellent paper that adds significantly to the literature to inform our understanding of the issues that junior doctors face. This is especially true because it is a qualitative

	paper that captures a more nuanced understanding of the issues compared to the many quantitative studies. These nuanced understandings assist when crafting appropriate interventions. Once these issues are addressed, it will be a significant addition to the doctors' health literature.
--	---

VERSION 1 – AUTHOR RESPONSE

Reviewer: 1

Reviewer Name: Miharu Nakanishi

Reviewer: 2

Reviewer Name: Andrea Cioffi

Reviewer: 3

Reviewer Name: Margaret Kay

Reviewer: 1

Institution and Country: Tokyo Metropolitan Institute of Medical Science, Japan

Reviewer: 2

Institution and Country: Sapienza University of Rome, Italy

Reviewer: 3

Institution and Country: University of Queensland, Australia

Reviewer: 1

Comments to the Author

The present study examined protective factors and sources of support in workplace among foundation and junior doctors who experienced stress, distress, anxiety, depression and suicidal thoughts.

The authors clearly summarised the core components from narratives. It would be recommended to discuss possible facilitators and barriers in NHS - as this study emphasised the scarce evidence in organisational culture and systemic factors - to establish peer support and supportive leadership strategies in the workplace.

Thank you for this positive comment – we have now included a statement on key workplace barriers and facilitators to help-seeking, previously reported in the literature (p.7/8).

As participants mentioned in the narratives, these factors may have varied across different workplaces. It thus seems to depend on some personal attributes of leaders whether the workplace can establish supportive culture and strategies. Does NHS have had specific strategy to reduce such dependency on individual leader's skills? Further discussion would be helpful for international readers to provide implications for healthy workplace for young doctors.

Thank you for this comment – we have now discussed the limitations of over-depending on leadership skills and qualities and incorporated this within the relevant discussion point (p.16).

Discussion on access to professional support, with citation of PHP, would also be beneficial.

Participants stated that they preferred such services rather than mental health services at local trust, because of accessibility or their concerns about being acknowledged by possible colleagues in the future. The latter would be worthy of note, as it can be specific to foundation and junior doctors.

This is discussed on page 15.

Reviewer: 2

Comments to the Author

Interesting article. The topic chosen is extremely topical and the typical problems of junior doctors are particularly important today because they influence the performance of the National Health Systems. The Manuscript is well structured and methodologically correct. Thank you for this positive comment.

It is advisable to report the first part of the findings section ("Twenty-one interviews were conducted") in the methods section. This because the research methods are described: number of interviews, interview time, questionnaire structure (3 themes).

Thank you for this comment. We would prefer to keep the sentence 'twenty-one interviews were conducted....' as we believe this does represent a finding. In addition, most manuscripts reporting qualitative studies will include the number of interviews conducted in the FINDINGS section of the paper.

Reviewer: 3

Comments to the Author

Thank you for the opportunity to review this paper "Protective factors and sources of support in the workplace as experienced by UK foundation and junior doctors: a qualitative study." The paper is well written and easy to read providing important practical understandings regarding the protective factors and sources of support for junior doctors. The authors would like to thank the reviewer for this positive comment.

The Abstract seems adequate as a summary of the article, though it would be helpful to have a little more information about the participants/recruitment here if possible. We have now included more information about participants and recruitment in the abstract (p.4).

In the Strengths and Limitations section, the authors could include the fact that the junior doctors who were recruited were doctors who had experienced mental health issues. This is significant because these doctors will visualise the issues through that lens and this is likely to impact their reporting of the issues being researched. It potentially limits the transferability of the findings across the junior doctor population. It can also be viewed as a strength. Either way it needs to be identified as something that could impact the findings.

We thank the reviewer for this comment and we have now included this within the Strengths and Limitations section (p.5).

The authors mention that they included many more women than men. While there were more women compared to men who were potential participants (37:22), the ratio of women to men in the actual participants after purposive sampling included a higher proportion of women (16:5). Given that gender was one of the issues considered in the purposive sampling, it is not really clear why there needed to be such significant weight given to including women. The justification in the Strengths and Limitations section is not adequate and this should be addressed clearly in the paper – perhaps in the methods section.

We thank the reviewer for this comment and we have now revised this limitation and provided greater clarity for the ratio of male and female participants. This has now been moved to the main body as this allows for a more in-depth explanation (p.16).

The introduction is a succinct description of the current literature that has a focus on the UK/Irish literature and a broader review of the literature is relevant here. The study is internationally relevant. The work of Shanafelt et al at Mayo Clinic on workplace culture and organisational aspects of doctor wellness could be included.

We are grateful for this comment and citation – we have now expanded the background section to incorporate relevant international literature.

While the authors are correct that there is a paucity of literature on junior doctors, there are studies, including Hoffman and Bonney (2018) “Junior doctors, burnout and wellbeing et al.” that reports findings that are relevant to this current study. While studies may focus on barriers, they may still report on findings about supportive work practices and engaging with the broader literature would strengthen this paper. Searching the literature again and including such papers would strengthen the introduction significantly.

Thank you for this comment and reference – we have now included Hoffman and Bonney’s work alongside relevant literature (p. 6)

On page 7 – lines 53-56 – the sentence, “Identifying such patterns or contextual factors is crucial in developing targeted improvements or interventions that can reduce the risk of mental ill health and psychological distress from the outset.” Is unclear in its meaning. Perhaps the authors could provide develop this statement to provide more detail for the reader who may not be as familiar with the workplace theories. It would be helpful to then refer to the more relevant theory/theories in their discussion to round out this concept.

We have now amended this sentence to make it clearer (we hope) and we discuss the importance of supportive work cultures, including positive relationships and learning environments which act as a buffer against work demands.

Currently the discussion seems to refer to this theory in the phrasing of the concepts but this is not overtly stated. The references in this introduction seem to be to the job-demand-resources theory its variations. Is this a key theory that the authors wish to highlight? Do they feel that their data confirm this as a useful theoretical approach to understanding their findings, or do they have mixed feelings about how their data fit this theory? Currently the introduction of the concept of workplace theories, leaves the reader hanging to hear the authors’ conclusion.

We thank the reviewer for bringing this to our attention and we have now discuss the relevance of the job-demand-resources theory in relation to our findings; namely, that key resources such as support from colleagues, quality of the relationships with supervisors/consultants and feeling valued can buffers the impact of those demands related to their work (e.g. long hours, workload)

The Method section – see comments above about the gender ratio. The definition of PPIE members was not included in the methods and it would be worth clarifying for the reader how the PPIE members could and did inform this study about junior doctors in more detail. It is possible that this contribution was significant and it is of interest to the reader. If it was not a significant contribution, then understanding how to enhance that contribution into the future for other researchers is equally important.

We have now moved the PPIE section to the methods section and provided more detail on how PPIE members shaped the need for this study (p.9).

Doctors’ health and wellbeing is an important issue for our community as a whole given the association between maintaining the wellbeing of doctors and the delivery of quality care.

I’m not 100% sure what point is being made here, but maybe refer to the part of the Discussion which refers to how doctors’ well-being is known to affect patient outcomes

We have now addressed this.

In the Method section, it seems to be implied that the analysis was inductive, but this needs to be clarified by the authors. If this was not the case, more detail would be helpful. It would also be useful to know the number of face-to-face and the number of the other type of interviews that were undertaken (p.8).

The analysis was inductive and we have now made this explicit. We have added the numbers of interviews conducted face-to-face, by telephone and using a digital platform (p.8).

In the results section, the definition of peer support seems quite broad. It seems that the quotes include the support from nurses and other team members, not just peers. The authors may wish to reconsider the title of this key theme that better encapsulates this breadth of support received by the junior doctors e.g. such as support by work colleagues.

We are grateful for this observation and the study team agree. We have now changed this theme and references to it: Support from work colleagues

The Discussion is well presented. (see notes about the Introduction above regarding the introduction of the workplace theory). It would benefit from a deeper engagement with the international literature on this topic, especially the other qualitative studies. There are also advantages to situating the qualitative data within the known quantitative data about junior doctors. Currently the discussion seems to focus on NHS and healthcare management references rather than references to the doctors' health literature. The findings are reinforced as strong statements but should be strengthened with reference to the existing knowledge about doctors' health.

Thank you for this comment – the authors have now included more international literature in the Introduction and Discussion sections, to ensure that the findings of this paper are pertinent to international readers. We have also included both quantitative and qualitative evidence, including literature on the mental health of doctors.

On page 15, the abbreviation of PHP is used in line 21 before it is explained in line 23. This has now been addressed.

The PHP is mentioned in the discussion. It is not clear if the authors think that this is the best option, rather than a closer GP? Accessing health care with a trusted GP is likely to be a reasonable option and perhaps this should be highlighted in the discussion as well?

Thank you and we agree and we have now added trusted GPs as a reasonable option for doctors seeking help.

The discussion includes some recommendations to address the concerns raised, but they seem to be quite vague concepts. Is it possible for the authors to give more specific recommendations?

The authors have included more specific recommendations, for example, drawing on the evidence base and practical recommendations for peer support as a buffer against mental ill health, particularly within a supportive learning/working environment and following critical incidents (p.14-16)

The references seem appropriate, but predominantly have a narrow focus on UK literature. A number of references, including 7, 10,11, 12,13,19,23,26,28,35,37, seems incomplete.

References have been updated; many of those listed above are reports.

Overall, I think this is an excellent paper that adds significantly to the literature to inform our understanding of the issues that junior doctors face. This is especially true because it is a qualitative paper that captures a more nuanced understanding of the issues compared to the many quantitative studies. These nuanced understandings assist when crafting appropriate interventions. Once these issues are addressed, it will be a significant addition to the doctors' health literature.

We are grateful to the reviewer for these positive comments and we feel that the paper has been strengthened by addressing these constructive comments.

VERSION 2 – REVIEW

REVIEWER	Andrea Cioffi Sapienza University of Rome, Italy
REVIEW RETURNED	25-Feb-2021

GENERAL COMMENTS	The revised manuscript is acceptable without further revision
---

REVIEWER	Dr Margaret Kay University of Queensland
REVIEW RETURNED	07-Mar-2021

GENERAL COMMENTS	It was a pleasure to read this revised article on such a very important topic. While there are many articles documenting the severity of burnout and other mental health issues in junior doctors, this paper provides qualitative data that enables a deeper understanding and considered the systemic issues that can contribute to the workplace both positively and negatively. It also demonstrates what supports junior doctors are looking for to address their health care needs. The paper is easy to read and logical and makes a significant and unique contribution to the current literature in this space. The introduction provides a comprehensive background for the study. The method is sound and consistent with the COREQ requirements. The results section includes relevant quotes that adequately illustrate the themes identified. The discussion is comprehensive and integrates the research findings with the current literature and offers important insights that will have important implications for enabling better support for junior doctors within the workplace as well as treatment options into the future. It is extensively and appropriately referenced. Reference 13 needs an organisation and weblink. The references are well chosen. I wonder if references 1, 3, 9, 10, 22, 25, 49, and 52 probably require a weblink. Reference 32 needs a correction – removing Elsevier and adding the volume and page numbers Reference 33 needs correction – I think that the 2009 version was the 2nd edition and I don't think there needs to be a volume number as this is a book. Reference 35 needs more information as it currently contains just the weblink Lines 33-36 on Page 15 have some inconsistent type. I think that the COREQ statement needs to be updated for the current revised submission, but the issues seem to be addressed well with information complete, just the wrong page numbers now. I recommend that the paper be accepted for publication.
---